# Pre-disaster mapping with drones: an urban case study in Victoria, BC, Canada

Maja Kucharczyk, Chris H. Hugenholtz

Department of Geography, University of Calgary, Calgary, AB T2N 1N4, Canada

5 *Correspondence to*: Maja Kucharczyk (maja.kucharczyk@ucalgary.ca)

**Abstract.** We report a case study using drone-based imagery to develop a pre-disaster 3D map of downtown Victoria, British Columbia, Canada. This represents the first drone mapping mission over an urban area approved by Canada's aviation authority. The goal was to assess the quality of the pre-disaster 3D data in the context of geospatial accuracy and building representation. The images were acquired with a senseFly eBee Plus fixed-wing drone with real-time kinematic/post-processed 10 kinematic functionality. Results indicate that the spatial accuracies achieved with this drone would allow for sub-meter building collapse detection, but the non-gimbaled camera was insufficient for capturing building facades.

## 1 Introduction

### 1.1 Background

Currently, 55 % of the global population resides in urban areas, and this is projected to increase to 68 % by 2050 (United 15 Nations, 2018). Increasing global population and urbanization (particularly in vulnerable areas) are factors that can contribute to increased death and destruction by natural hazards like earthquakes and tropical cyclones. In addition to initiatives such as increased quality of construction, alarm systems, and proximity to rescue services, pre-disaster mapping can help increase a city's resilience to disasters (Pu, 2017). Maps combining vector layers, digital elevation models, and aerial/satellite imagery are powerful tools for mitigation and preparedness before a disaster strikes (Pu, 2017). Copernicus Emergency Management 20 Service (Copernicus EMS), one of the main contributors of disaster management maps globally, has used pre-disaster data to produce thousands of reference maps (showing territories and assets), pre-disaster situation maps (showing hazard levels, evacuation plans, and modeling scenarios), and damage grading maps (showing the distribution and level of damage to buildings and infrastructure). Copernicus EMS generates damage grading maps by visually comparing pre- and post-disaster satellite imagery (Copernicus EMS, 2017). Nearby debris is used as a proxy for building structural damage, as building facades 25 cannot be directly examined (Copernicus EMS, 2017).

The leading cause of death in an earthquake is building collapse (Moya et al., 2018). Remote sensing can potentially assist first responders in rapidly locating collapsed buildings to prioritize search and rescue efforts (Moya et al., 2018). However, clouds and access to satellite imagery can cause delays in analysis and preclude the traditional 2D approach from being useful for search and rescue. Furthermore, partial building collapse, which can trap and kill victims, generates lower

amounts of debris than complete collapse, so the dependence on debris as a proxy for collapse becomes less reliable. Research has shown that 2.5D data can build upon the traditional 2D approach and increase the reliability of collapse detection by observing elevation changes over buildings. Following the 2016 Kumamoto earthquake in Japan, Moya et al. (2018) detected collapsed buildings using pre- and post-earthquake light detection and ranging (LiDAR) digital surface models (DSMs). For each building, they calculated the average height difference between the DSMs and manually set a threshold value to detect collapse – this technique had a Cohen's kappa coefficient and overall accuracy of 0.80 and 93 %, respectively (Moya et al., 2018). Pre-event LiDAR data, however, can often be outdated, leading to false detections, or unavailable, especially in less-developed parts of the world. Post-event LiDAR data may be difficult to rapidly obtain. To address these operational challenges, drones are an alternative platform for acquiring 2.5D and 3D data, and when stored locally for emergency mapping, can be used to rapidly acquire data. Drone-derived aerial imagery, when paired with structure-from-motion multiview-stereo image processing software, can be used to generate sub-decimeter resolution orthomosaics, DSMs, and photorealistic 3D models in the form of colorized point clouds and textured meshes.

Drone-based mapping can also potentially support longer-term needs assessments and reconstruction monitoring by surveying building damage levels. The traditional 2D approach with satellite imagery only provides information about building roofs and nearby debris, and previous research has shown that oblique perspectives of building facades are valuable for discerning between lower grades of building damage (Kakooei and Baleghi, 2017; Masi et al., 2017). Previous studies have conducted drone-based 3D mapping of buildings following a disaster. The motivation is to complement ground-based building damage assessments – cataloging the exterior damages in 3D can support the planning/prioritizing of subsequent, more-thorough ground-based assessments (Vetrivel et al., 2018), and the planning/monitoring of reconstruction. Previous studies (e.g., Fernandez Galarreta et al., 2015; Cusicanqui et al., 2018) have reported that damage features such as deformations, cracks, debris, inclined walls, and partially collapsed roofs are identifiable in drone-based 3D point clouds and mesh models. These findings demonstrate that drone 3D data are capable of supporting post-disaster activities. However, previous studies have been limited to drone-based 3D mapping: (i) a single building (Achille et al., 2015; Meyer et al., 2015), (ii) small, historic villages (Vetrivel et al., 2015; Dominici et al., 2017; Calantropio et al., 2018; Cusicanqui et al., 2018; Vetrivel et al., 2018), or (iii) modern cities, but without focus on the quality of building representation in the 3D data (Cusicanqui et al., 2018; Vetrivel et al., 2018). It is important to understand how drone-based 3D data would reconstruct a cityscape, particularly with a grid-based survey to capture multiple city blocks in a single flight. This flight pattern would balance areal coverage with 3D reconstruction quality. The dense spacing of buildings and the presence of high-rises in an urban scene create considerable potential for camera occlusion and may result in 3D mesh defects such as inaccurate shapes, holes, and blurred textures (Wu et al., 2018).

In addition to issues with photogrammetry, it is challenging to collect drone data over dense, urban areas due to aviation regulations that were designed to protect public safety. As such, in a disaster context, drone data over cities have generally been collected in the post-disaster phases, when destruction is widespread and these data are in high demand. With historic emphasis on data collection in the post-disaster phases, it is important to not detract from pre-disaster mapping. Pre-

disaster mapping not only provides baseline data from which to assess changes, but is also a crucial exercise that enables emergency management actors to establish operational protocols to maximize the effectiveness of drones in emergencies. These protocols pertain to drone hardware/software, data collection, data processing, and data analysis.

We present a case study of pre-disaster mapping with a drone in Victoria, British Columbia, Canada. Victoria has at
least a 30 % probability of experiencing a significantly damaging earthquake in the next 50 years (AIR Worldwide, 2013). A 2016 report on the seismic vulnerability of Victoria presented a risk assessment for all buildings (13,330 buildings) in Victoria under various earthquake scenarios and levels of ground shaking (VC Structural Dynamics Ltd, 2016). The report concluded that 30 % of the buildings (3,936 buildings) have a high seismic risk, meaning they have at least a 5 % probability of complete damage in a 50-year period (VC Structural Dynamics Ltd, 2016). This pre-disaster mapping exercise was undertaken for City
of Victoria's Emergency Management Division and in partnership with GlobalMedic, a Canadian disaster relief charity. This was the first Transport Canada-approved drone mapping mission over a major Canadian city. We were restricted by regulations to use a specific platform, a 1.1 kg senseFly eBee fixed-wing drone. The overarching goal of this case study was to assess the quality of the drone data that we were able to obtain in a manner adhering to federal regulations.

## 1.2 Objectives

The first objective was to assess the geospatial accuracy of the drone data. Geospatial accuracy is important for change detection applications, as it relates to the quality of registration between pre- and post-disaster datasets. This was done by first assessing the vertical accuracy of the drone DSM using 47 ground-surveyed checkpoints. Then, a LiDAR DSM was subtracted from the drone DSM to visually assess the horizontal alignment of rooftops as a qualitative measure of horizontal accuracy. The second objective was to assess the quality of 3D building representation. The only drone legally approved for urban
overflight in Canada in 2018 presents challenges for 3D mapping of cities, as it is a fixed-wing drone with a non-gimbaled camera. Research has shown that high camera tilt angles, which are not achievable with the regulatory platform for this flight, will result in higher reconstruction density (less data gaps) and precision of points on building facades than lower camera tilt angles (Rupnik et al. 2015). The quality assessment of 3D building representation was done by visually assessing the drone 3D textured mesh, and using Google 3D (i.e., "3D Buildings" layer in Google Earth) as a reference for building appearance.
Additionally, we applied a method previously used on post-disaster, drone-derived 3D point clouds to quantify data gaps on sample building facades.

## 1.3 Regulatory background

Transport Canada is the aviation authority that regulates drone operations in Canadian airspace. The regulations in 2018 required case-by-case permission for drone flights in urban areas. Permission was sought by submitting an application for a
Special Flight Operations Certificate. This application demonstrated sufficient ground/flight training, standard operating procedures, emergency procedures, drone maintenance procedures, and more. Additionally, coordination with air traffic control (Nav Canada) was required to perform the flight, as downtown Victoria is within controlled airspace, with nearby

airports, heliports, and seaplane bases causing high-density air traffic. In 2018, the only drone legally approved for urban overflight in Canada was the senseFly eBee, of which the "Plus" model was used for its higher georeferencing accuracy. The senseFly eBee Plus is a 1.1 kg, 1.1 m wingspan, fixed-wing drone made of lightweight expanded polypropylene foam, carbon fiber, and composite materials. At the time of the flight, the eBee Classic, SQ, and Plus models were the lightest on the list of

compliant drones for Transport Canada, which included drones meeting federal safety and quality standards. For this flight, the senseFly eBee drone was approved by Transport Canada due to its light weight and ability to glide to a landing.

## 2 Methods

### 2.1 Flight area

The drone flight covered a 1 km$^2$ area of downtown Victoria, BC, Canada. The western half and eastern half of the flight area

covered parts of the Historic Commercial District (HCD) and Central Business District (CBD), respectively, resulting in image capture over a diversity of building types and heights. The HCD contains an undulating streetscape with low- to mid-rise, brick- and stone-facade buildings alternating between one and five stories, including boutique hotels, heritage buildings, businesses, and offices (CoV, 2011). The CBD contains high-density, mid- to high-rise commercial and residential buildings (CoV, 2011). The building heights within the flight area ranged from 4–55 m, and street widths varied between 7–24 m.

### 2.2 Drone hardware and flight planning

A senseFly eBee Plus drone with real-time kinematic (RTK)/post-processed kinematic (PPK) functionality and senseFly Sensor Optimised for Drone Applications (SODA) red-green-blue (RGB) 20-megapixel camera were used to collect imagery. The RTK/PPK image georeferencing capabilities of the drone replaced the need for ground control points (GCPs), which are not practical to distribute and survey in an emergency (i.e., post-disaster) mapping context. It is important to note that, for pre-

disaster mapping, GCPs should be used to maximize geospatial accuracy. Hugenholtz et al. (2016) demonstrated the improvement in DSM vertical accuracy when using a non-RTK/PPK senseFly eBee with GCPs compared to an RTK/PPK-enabled senseFly eBee without GCPs. For this pre-disaster mapping exercise in downtown Victoria, we chose to use RTK/PPK image georeferencing because this method is also applicable to post-disaster mapping. One of our objectives was to assess the geospatial accuracy of the pre-disaster data, which has implications for the use of RTK/PPK-enabled drones for post-disaster

mapping and change detection applications.

The drone's PPK mode was used, with correction data obtained from the Natural Resources Canada (NRCan) Canadian Active Control System (Albert Head reference station, 10 km from flight area). SenseFly eMotion (v3) software (senseFly, 2018) was used to plan the flight. The flight was grid-based, composed of orthogonal flight lines running non-parallel with streets (i.e., approximately 45 ° offset). The addition of perpendicular flight lines and the orientation of the grid

were used to increase image coverage of building facades. The imagery frontal and lateral overlap were set to 75 %, and the flight altitude was 120 m above ground level (AGL).

## 2.3 Drone image acquisition

The flight was conducted in the morning hours of June 14, 2018. A morning flight was chosen to coincide with low air traffic; however, the tradeoff for image acquisition is increased building shadow area and data occlusion compared to solar noon. The ground control station was set up on a parkade rooftop within the flight area. The parkade, surrounded by relatively low buildings and an open courtyard, allowed for unobstructed takeoff/landing, visual line of sight, and radio signal between the drone and ground control station. A total of 828 images were captured. The median image pitch angle was 7.35 ° off nadir (3.55 ° interquartile range), with a minimum and maximum of 1.22 ° and 11.83 °, respectively.

## 2.4 Image processing

The images were processed using a high-performance computer (Intel® Core™ i9-7900X CPU @ 3.30 GHz with 64 GB RAM and NVIDIA GeForce GTX 1080 GPU). First, senseFly eMotion software was used for PPK processing by incorporating raw global navigation satellite system (GNSS) observations from the reference station and drone to refine the image geotags. The geotagged images were processed using Pix4Dmapper Pro (v4.3.27) (Pix4D) (Pix4D, 2018), which is a structure-from-motion multiview-stereo (SfM-MVS) software. SfM-MVS generally consists of the following steps (as outlined by Westoby et al., 2012). First, computer vision algorithms search through each image to identify "features", which are pixel sets that are robust to changes in scale, illumination, and 3D viewing angle. Next, the features are assigned unique "descriptors", which allow for the same features to be identified across multiple images, and for the images to be approximately aligned. This initial image alignment is iteratively optimized via bundle adjustment algorithms, the output of which is a sparse 3D point cloud of feature correspondences. Multiview-stereo algorithms then densify the sparse point cloud, typically by two or more orders of magnitude. The dense point cloud is then used to generate a 3D textured mesh, which is a triangulated surface that is textured using the original images. The dense point cloud is also used to generate a DSM. The DSM and images are used to generate an orthomosaic.

For the first objective of assessing the geospatial accuracy of the drone data in an urban context, five DSMs were generated. Each DSM had increasingly computationally intensive parameters, resulting in an increasingly higher processing time, ranging from 0.50–8.14 h. This has important implications on the applicability of a drone-based DSM for rapid building collapse detection, where time is a major factor for emergency responders. Four "rapid" DSMs were generated in Pix4D using values of 1/8, 1/4, 1/2, and 1 for the image scale parameters (Step 1: keypoints image scale and Step 2: image scale), and low density for the point cloud. One "slow" DSM was generated using a value of 1 for the image scale parameters, and optimal (medium) density for the point cloud. All 5 DSMs were generated using 3 minimum matches, noise filtering, "sharp" surface smoothing, and inverse distance weighting interpolation. For the second objective of assessing 3D building representation, a 3D textured mesh was generated in Pix4D using a value of 1 for the image scale parameters, optimal (medium) density for the point cloud, 3 minimum number of matches, and high resolution for the textured mesh. A medium-resolution mesh was also generated for comparison to the high-resolution mesh.

All Pix4D data outputs had a spatial reference of Universal Transverse Mercator (UTM) Zone 10N, North American Datum of 1983 (Canadian Spatial Reference System) (NAD83 [CSRS]), using the Canadian Geodetic Vertical Datum of 2013 (CGVD2013) for orthometric heights relative to the Canadian Gravimetric Geoid model of 2013 (CGG2013, 2010.0 epoch). The DSMs were assessed for geospatial accuracy, while the point cloud and textured meshes were used to assess 3D building

representation in terms of geometry and texture.

## 2.5 Geospatial accuracy assessment

To be useful for change detection, such as generating a DSM of difference (DoD) for building collapse detection (e.g., Moya et al., 2018), the drone data must be geospatially accurate. Otherwise, misregistration of the drone data with pre- or post-event data may cause false detections. Therefore, the vertical accuracy of each drone DSM was assessed using 47 ground-surveyed

checkpoints. The vertical accuracy assessment was conducted using recommendations from the 2015 American Society for Photogrammetry and Remote Sensing (ASPRS) Positional Accuracy Standards for Digital Geospatial Data (ASPRS, 2015). ASPRS (2015) recommend vertical checkpoints to be ground-surveyed and located on flat or uniformly sloped ($\leq 10$ % slope), open terrain, away from vertical artifacts and abrupt elevation changes. The checkpoints used in this accuracy assessment were collected using a total station, and represent sewer manhole covers located on paved roads throughout the study area. Each

checkpoint $z$-coordinate ($z_{ref}$) was subtracted from the corresponding drone DSM value ($z_{drone}$) to calculate errors ($z_{drone} - z_{ref}$). A Shapiro-Wilk test ($\alpha$ level of 0.05) and a visual inspection of the histogram, normal Q-Q plot, and box plot indicated that the errors followed a normal distribution. Therefore, vertical accuracy was calculated as the vertical root mean squared error (RMSE$_z$) following Eq. (1):

$$RMSE_z = \sqrt{\frac{1}{n}\sum_{i=1}^{n}(z_{i(drone)} - z_{i(ref)})^2}, \qquad\qquad (1)$$

where $z_{i(drone)}$ is the value of the $i$th cell from the drone DSM, $z_{i(ref)}$ is the $z$-coordinate of the corresponding checkpoint, and the total number of observations is represented by $n$ (ASPRS, 2015). To assess the implications of the vertical accuracies, a level of detection (LoD) was calculated to determine the threshold elevation difference that can be detected using pre- and post-disaster DSMs with known RMSE$_z$ values, following Eq. (2):

$$LoD = \pm 3 \times \sqrt{(RMSE_{z1})^2 + (RMSE_{z2})^2}, \qquad\qquad (2)$$

where $RMSE_{z1}$ is the RMSE of the pre-disaster DSM, $RMSE_{z2}$ is the RMSE of the post-disaster DSM, and the multiplier, 3, represents the extreme tails of a normal probability distribution (Hugenholtz et al., 2013). The LoDs for several hypothetical DoDs were calculated. Each hypothetical DoD contained a different combination of pre- and post-disaster DSMs. The DSMs included the most "rapid" PPK-corrected drone DSM (processed in 0.50 h), the "slow" PPK-corrected drone DSM (processed in 8.14 h), a non-RTK/PPK drone DSM, and a LiDAR DSM. The RMSE$_z$ values for the PPK-corrected drone DSMs were

experimentally derived in this study. The RMSE$_z$ value for the non-RTK/PPK drone DSM was experimentally derived by Hugenholtz et al. (2016). Based on 180 RTK GNSS vertical checkpoints from a gravel pit, Hugenholtz et al. (2016) calculated an RMSE$_z$ of 2.144 m for a non-RTK/PPK senseFly eBee (no GCPs). The RMSE$_z$ value for the LiDAR DSM was

experimentally derived in this study. The LiDAR data were acquired in 2013 with a Leica ALS70-HP sensor from an average flight altitude and speed of 1360 m AGL and 220 knots, respectively. The field of view and average swath width were 47 ° and 1240 m, respectively. The scan rate was 48.9 Hz, and the laser pulse rate was 370.6 kHz. The LiDAR point cloud was interpolated into a 0.31 m DSM in ESRI ArcMap (v10.5.1) (ESRI, 2018) using inverse distance weighting interpolation and

linear void fill. Using the 47 ground-surveyed checkpoints, the $RMSE_z$ of the LiDAR DSM was calculated.

To visually assess the horizontal accuracy of the drone data, a DoD was generated by subtracting the LiDAR DSM from the "slow" PPK-corrected drone DSM. The DoD was used to visually assess the horizontal alignment of roofs as a qualitative measure of horizontal accuracy.

## 2.6 Assessment of building geometry and texture

In addition to geospatial accuracy, we assessed the quality of building representation in the drone-derived 3D data. This assessment has implications on the usability of the 3D data for identifying damages to building roofs and facades. The medium- and high-resolution textured meshes were visually assessed for quality of building representation in terms of geometry and texture. Eight sample buildings ranging in geometrical complexity were segmented from each mesh using CloudCompare (v2.9.1) (CloudCompare, 2018) and were visually compared. Google 3D (i.e., Google Earth layer "3D Buildings") served as

a reference for building appearance (Google, 2018). The Google 3D layer was photogrammetrically derived using nadir and 45 ° aerial imagery that was collected with a multi-camera system in 2014. To support the visual assessment, each sample building was segmented from the dense point cloud, and each building point cloud was colored by 3D point density using CloudCompare. To further investigate geometrical and textural distortions within the mesh, the dense point cloud was used to quantify data gaps on building facades (i.e., regions of facades without points). The procedure generally followed Cusicanqui

et al. (2018), who assessed the completeness of drone-based point clouds of post-earthquake study areas in Taiwan and Italy. Using CloudCompare, six sample facades were segmented from the dense point cloud. The Rasterize tool was used to project the points of each segmented facade onto a 0.50 m grid, with the projection plane parallel to the facade. Then, a 0.50 m raster was generated, showing the number of 3D points in each cell. For each raster, the percentage of facade data gaps was calculated by dividing the number of empty cells by the total number of cells. To support the data gap assessment, the sample facades

were also segmented from the high-resolution mesh using CloudCompare.

## 3 Results

### 3.1 Geospatial accuracy of drone DSM

The vertical errors of the PPK-corrected drone DSMs and the LiDAR DSM were calculated using 47 ground-surveyed checkpoints located on sewer manhole covers. For the "slow" PPK-corrected drone DSM, errors ranged from 0.03–0.13 m

(Fig. 1a). The mean vertical error was 0.08 m, with a standard deviation of 0.02 m (Fig. 1a), demonstrating that the drone DSM tended to overestimate the elevation of the ground surface. Table 1 shows the $RMSE_z$ values of the "slow" PPK-corrected

drone DSM and 4 "rapid" PPK-corrected drone DSMs. $RMSE_z$ decreased as total processing time increased (Table 1). The DSM generated in the least amount of time, 0.50 h, had an $RMSE_z$ of 0.16 m, which is 0.08 m higher than the $RMSE_z$ for the "slow" drone DSM, generated in 8.14 h (Table 1). The non-RTK/PPK drone DSM and LiDAR DSM had $RMSE_z$ values of 2.144 m and 0.04 m, respectively.

5       Table 2 shows hypothetical DoDs, each generated with a different combination of pre- and post-disaster DSMs, and their resulting LoDs from Eq. (2). For each hypothetical DoD, the corresponding LoD value indicates that any elevation difference between -LoD and +LoD is likely due to error and cannot be interpreted as real. DoDs generated with one or more DSMs derived from a non-RTK/PPK drone (DoD5 and DoD6) had LoDs of 6.43 m and 9.10 m (Table 2). For LoDs attributed to the use of non-RTK/PPK drones, buildings shorter than the LoDs cannot be assessed for collapse, and for assessable

buildings, only DoD values exceeding the LoDs are likely to correspond to real collapse. The 6.43 m and 9.10 m LoDs exceed the typical height of a single building story, suggesting that DoDs generated with non-RTK/PPK drones cannot be reliably used to detect partial collapse. Conversely, DoDs generated with one or more DSMs from an RTK/PPK drone (DoD1–4) had LoDs ranging from 0.27–0.54 m, suggesting that these DoDs can be reliably used to detect partial collapse (Table 2). This includes DoD3 and DoD4, both generated using a "rapid" PPK-corrected post-event drone DSM (Table 2). The use of the

lowest image scale value (1/8) and lowest point density in Pix4D to generate the most rapid DSM in 0.50 h (Table 1) retained sub-meter LoDs (0.49–0.54 m) (Table 2).

       The horizontal accuracy of the "slow" PPK-corrected drone DSM was visually assessed by calculating a DoD with the LiDAR DSM (Fig. 1b). The DoD shows blue tints for elevation overestimations and red tints for elevation underestimations by the drone DSM (Fig. 1b). A 2013 satellite image was viewed in Google Earth and compared to the drone orthomosaic to

determine buildings common to both datasets. Figure 1b identifies 16 buildings with large regions of contiguous DSM differences. These contiguous DSM differences are due to changes that occurred between the 2013 LiDAR and 2018 drone data acquisitions, such as new construction, structure removal, and parking lot excavation (Fig. 1b). For the rest of the DoD, the red and blue cells mostly correspond to changes in vegetation and inconsistencies in building footprint edges between the drone and LiDAR DSMs (Fig. 1b). Building outlines appear mostly blue and don't appear weighted more heavily in one

direction (Fig. 1b), suggesting no major horizontal offset of the drone DSM relative to the LiDAR DSM. With a 0.31 m average point spacing, it is possible that the LiDAR point cloud did not sample roof edges, resulting in slightly smaller building footprints in the LiDAR DSM than the drone DSM. Building footprint edge differences could also be due to inaccurate geometry from drone-based photogrammetry.

## 3.2 Building representation: mesh resolution and data gap assessment

The appearance of buildings varied considerably between the medium- and high-resolution 3D meshes. Figure 2 shows eight sample buildings represented by the dense point cloud (colored by 3D point density), both meshes, and Google 3D as a reference. Both meshes were generated using the settings described in § 2.4, with only the mesh resolution setting varying. For each building, the point density is higher on roofs than facades, and data gaps (i.e., regions of zero points) are visible within

facades (Fig. 2). The medium-resolution mesh has visibly poorer reconstruction of building geometry and, subsequently, more deformations in texture than the high-resolution mesh (Fig. 2). This was expected, as each medium-resolution building contains only 4–5 % of the vertices/faces of its high-resolution counterpart. Figures 2a–2d show heritage buildings with complex geometry: Victoria City Hall (Fig. 2a), St. John the Divine Anglican Church (Fig. 2b), Alix Goolden Performance Hall (Fig. 2c), and St. Andrew's Cathedral (Fig. 2d). Smaller architectural features common to these heritage buildings, such as gabled entrances, dormer windows, conical roofs, spires, and towers are better resolved in the high-resolution mesh (Fig. 2a–d). For these buildings, as well as buildings with simpler geometry (Fig. 2e–h), the high-resolution mesh shows higher linearity of facade, roof, and window edges. For the high-rise buildings (Fig. 2e–h), facades with widespread data gaps in the point cloud appear to protrude inward and outward in the meshes, and have severe textural distortions. For generally planar facades with regular sampling (e.g., the front-facing facades in Fig. 2e and 2f), the apparent geometrical and textural differences between the medium- and high-resolution meshes are less prominent. The 95–96 % lower density of vertices/faces in the medium-resolution mesh appears more robust to geometrical/textural distortions for buildings with simpler, planar geometry than those with complex geometry, provided there is adequate sampling. However, as demonstrated by the high-rise buildings (Fig. 2e-h), facades with widespread data gaps have severe distortions, regardless of mesh resolution.

The point clouds in Fig. 2 show that roofs were more densely and regularly sampled than facades, and some facades contained widespread gaps that resulted in severe distortions in the meshes. To further assess facade data gaps, particularly partial data gaps, six facades were segmented from the dense point cloud and high-resolution mesh. The 0.50 m point density raster and high-resolution mesh segmentation are shown for each facade in Fig. 3. Data gaps, represented by red cells, encompass 9–59 % of the facades (Fig. 3). For each facade, large regions of contiguous red cells in the point density raster appear attributed to distortions in the mesh (i.e., stretched texture and inwardly protruding geometry).

## 4 Discussion

### 4.1 Key lessons: drone geospatial accuracy and up-to-date pre-disaster DSMs

For building collapse detection (e.g., Moya et al., 2018), drones can provide post-event DSMs that can be differenced with pre-event DSMs derived from LiDAR and other methods. However, there are geospatial accuracy requirements to minimize errors caused by the misregistration of pre- and post-event DSMs. As such, we conducted a vertical accuracy assessment of each PPK-corrected drone DSM and the LiDAR DSM using 47 ground-surveyed checkpoints. The $RMSE_z$ values were then used to calculate LoDs for hypothetical DoDs generated from different combinations of pre- and post-disaster DSMs. For DoDs attributed to the use of non-RTK/PPK drones, the LoDs exceeded the typical height of a single building story, suggesting that DoDs generated with non-RTK/PPK drones cannot be reliably used to detect partial collapse (Table 2). Conversely, DoDs generated with one or more DSMs from an RTK/PPK drone (DoD1–4) had LoDs ranging from 0.27–0.54 m, suggesting that these DoDs can be reliably used to detect partial collapse (Table 2). These results demonstrate that, in the absence of GCPs, RTK/PPK-enabled drones are required for reliable building collapse detection, and rapid processing settings can be used.

Furthermore, the DoD calculated by differencing the LiDAR DSM and "slow" PPK-corrected drone DSM showed buildings with large regions of contiguous DSM differences due to changes that occurred between the 2013 LiDAR and 2018 drone data acquisitions, such as new construction, structure removal, and parking lot excavation (Fig. 1b). This demonstrates that, for building collapse detection, it is necessary to maintain an up-to-date pre-disaster DSM to avoid false detections. Victoria, like most urban areas, has a dynamic downtown core with rezoning and new building construction or renovations to help accommodate significant population growth forecasted over the next 20 years (CoV, 2011). Routine updates to urban DSMs could help discriminate real, disaster-induced changes to buildings from those associated with construction or renovation activities. Up-to-date DSMs can be derived from many methods, including airborne photogrammetry and laser scanning. To reduce costs and time associated with continuously updating a pre-disaster DSM, future research should focus on developing methodologies to facilitate distinction between construction/renovation-modified and disaster-damaged buildings in a DoD.

## 4.2 Key lessons: drone mesh resolution and imaging platform

From an image processing standpoint, it was shown in § 3.2 that mesh geometry and texture were improved considerably from a medium-resolution to a high-resolution mesh (Fig. 2). The high-resolution mesh required more processing time, including subsetting the project into two, but these improvements justify the added time for virtual 3D damage assessment applications. For image collection, the deformed geometry and texture of buildings (Fig. 2 and 3) could be improved by collecting highly oblique images of facades in addition to nadir images of roofs and ground features. The median 7 ° camera pitch angle used in this case study was likely insufficient for capturing vertical and near-vertical faces, resulting in large point cloud data gaps and geometrical/textural distortions in the 3D mesh that could be mistaken for damage (Fig. 2 and 3). Using a higher camera angle (e.g., 30–45 ° off nadir) could make important improvements. Rupnik et al. (2015) found that increasing the camera tilt angle resulted in a higher point density on building facades and higher 3D precision of points, and that the addition of oblique images to a nadir image set increased the vertical accuracy of points. This suggests that different hardware is required for mapping municipalities in 3D with small drones. Options include multi-rotor drones with gimbaled cameras that are capable of highly oblique image capture. However, we suspect multi-rotor drones may be more difficult for urban overflight than fixed-wing drones from a regulation perspective. Most small multi-rotor drones have fixed-pitch rotors, which, in the event of power loss, do not allow for the drone to enter autorotation to substantially slow the descent (Perritt & Sprague, 2017). Fixed-wing drones, on the other hand, are able to glide after losing power (Perritt & Sprague, 2017). Therefore, a potential solution is to use a lightweight fixed-wing drone with a camera that tilts for oblique image capture. One commercially available option is the senseFly eBee X drone with a senseFly SODA 3D RGB camera, which captures one nadir image and two laterally oblique images per image waypoint. The eBee X model became available in September 2018, after the data capture in this study. Lightweight RTK/PPK-enabled multi-rotor drones may be more affordable than the senseFly eBee X with SODA 3D camera, but typically have a shorter battery life and subsequently lower areal coverage than fixed-wing drones.

It is important to note that a higher camera angle is not a panacea – higher camera tilt angles result in higher occlusions due to surrounding buildings, which contribute to lower point density on lower parts of facades (Rupnik et al., 2015). Moreover, point cloud gaps will persist on facades due to several factors, including: (i) occlusions caused by surrounding buildings, facade protrusions, and other objects; (ii) insufficient texture; (iii) highly reflective surfaces like glass; and (iv) poor image quality (Fonstad et al., 2013; Alsadik et al., 2014). Another potential solution is to obtain images of building facades from the ground. Wu et al. (2018) showed that drone-derived textured meshes of urban study areas in Germany and Hong Kong were improved with the integration of ground-based images. The meshes had increased geometric accuracy and improved texture (Wu et al., 2018). However, potential challenges in obtaining terrestrial images include added time, safety concerns, and limited access, especially in a post-disaster context.

## 4.3 Applicability to other urban areas

The data acquisition methods used in this study will need to be adapted to fit the conditions of different urban areas. For example, flight altitude will need to be adjusted to give a safe vertical clearance from the tallest building. If the terrain in the area is sloped, elevation data should be input to the flight planning software to keep the flight altitude constant. A grid of flight lines is recommended, although its orientation and image overlap will vary depending on factors such as building layout and density. Cities with taller buildings and greater variation in building height than Victoria may be difficult to map in 3D with a drone, as the high flight altitude may preclude full capture of facades. In a post-disaster context, a takeoff and landing location may be difficult to locate and access due to widespread destruction. Weather conditions such as high winds and rain following storm events may pose challenges to the flying ability of lightweight drones. Atmospheric conditions such as haze and smoke also limit optical sensors in imaging destruction.

The applicability of this study to other urban areas may be further challenged by legal restrictions in different national airspaces. For instance, a 2017 evaluation of drone regulations in 19 countries showed that 12 countries (including Canada, Italy, and China) prohibited drone flights over urban areas, and some countries also required minimum horizontal distances to be kept from these areas (Stöcker et al., 2017). Furthermore, 17 of 19 countries specified maximum flight altitudes, which ranged from 90–152 m AGL (Stöcker et al., 2017). Additional operational limitations include minimum horizontal distances from airports and heliports (Stöcker et al., 2017). For urban overflight outside of the regulatory operational limitations, special approval could be an option, although it is not guaranteed. Special approval would be required, for example, in cities where building heights exceed the maximum flight altitude, or cities with interior airports or heliports, which is common in most major urban areas. As in this case study, special approval and multi-stakeholder coordination are likely requirements for drone-based mapping in other urban areas. In summary, the applicability of this study to other cities will depend on the urban topography, weather and atmospheric conditions, and drone regulations.

## 5 Conclusions

We presented a case study of drone-based pre-disaster mapping in downtown Victoria, BC, Canada. The objectives were to assess the quality of the data in terms of geospatial accuracy and 3D building representation. Using 47 ground-surveyed checkpoints, the $RMSE_z$ of the drone-derived DSM was 0.08 m. The DSM of difference (DoD = $DSM_{drone}$ − $DSM_{LiDAR}$) showed
complete roof overlap, suggesting adequate horizontal accuracy for change detection applications. For building collapse detection, results suggest that drones with RTK/PPK image georeferencing capabilities and up-to-date pre-disaster DSMs are required to avoid false detections. Furthermore, image processing using "rapid" settings, as opposed to "slow" settings, reduced processing time from 8.14 h to 0.50 h, increased DSM $RMSE_z$ from 0.08 m to 0.16 m, and increased DoD LoD from 0.34 m to 0.54 m. Though processing times were specific to our computing hardware, these differences demonstrate that "rapid"
processing is capable of quickly generating DSMs that can reliably detect sub-meter building collapse. Conversely, hypothetical DoDs derived from one or more non-RTK/PPK drone DSMs have LoDs too high (i.e., > 6 m) to reliably detect partial building collapse. These results suggest that RTK/PPK-enabled drones and "rapid" image processing are most suitable for rapid building collapse detection with drones.

For building damage assessment with drone-derived 3D textured meshes, it was shown that a high-resolution mesh,
containing 95–96 % more vertices/faces than a medium-resolution mesh, visually improved building geometry and texture, especially for heritage buildings with complex geometries and small architectural features. However, neither mesh resolution was able to cope with large point cloud gaps on building facades. These data gaps were shown to correspond with severely distorted geometry and texture in the mesh. Therefore, for future drone-based pre- and post-disaster 3D mapping of municipalities, different hardware will be required. The ability to capture highly oblique images is paramount for
reconstructing building facades. Options include a multi-rotor drone with a gimbaled camera. However, lightweight multi-rotor drones may be more challenging for large-area mapping missions in major urban areas due to regulatory restrictions. Therefore, we suggest a follow-up study using a fixed-wing drone with oblique image acquisition capabilities.

### Data availability

The data used in this study are not publicly available due to privacy issues.

### Author contribution

CHH conceptualized the research goals, MK and CHH designed the methodology, MK executed the methodology, MK prepared the manuscript, and CHH reviewed and edited the manuscript.

**Competing interests**

The authors declare that they have no conflict of interest.

**Acknowledgements**

We would like to thank the City of Victoria for helping to coordinate the drone data acquisition and providing access to the
drone and LiDAR data, GlobalMedic for helping to design and coordinate the drone data acquisition, InDro Robotics and
Global UAV Technologies for acquiring the drone data, Nav Canada for coordinating the drone flights with the harbor control
tower and pilots, and Transport Canada for granting approval for the drone operations. We gratefully acknowledge the support
of Alberta Innovates and Alberta Advanced Education. We also appreciate the helpful comments and suggestions from the
anonymous reviewers and editor.

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

**Table 1. DSMs generated with different processing settings in Pix4D (v4.3.27), with processing time (minutes) for each step, total processing time (hours), and resulting RMSE$_z$. The processing was done using a high-performance computer (Intel® Core™ i9-7900X CPU @ 3.30 GHz with 64 GB RAM and NVIDIA GeForce GTX 1080 GPU).**

| DSM | Processing settings | | Processing time (min) | | | | |
| | Image scale | Point density | Initial processing | Point cloud densification | DSM generation | Total processing time (h) | DSM RSME$_z$ (m) |
| --- | --- | --- | --- | --- | --- | --- | --- |
| Rapid1 | 1/8 | Low | 8.97 | 9.78 | 11.15 | 0.50 | 0.16 |
| Rapid2 | 1/4 | Low | 13.07 | 13.60 | 10.80 | 0.62 | 0.14 |
| Rapid3 | 1/2 | Low | 22.15 | 28.62 | 13.57 | 1.07 | 0.11 |
| Rapid4 | 1 | Low | 25.73 | 105.22 | 28.08 | 2.65 | 0.08 |
| Slow | 1 | Medium | 25.83 | 361.00 | 101.80 | 8.14 | 0.08 |

**Table 2. Hypothetical DoDs calculated using different DSM combinations. The LoD for each DoD was calculated using Eq (2).**

| DoD | Pre-disaster DSM | Post-disaster DSM | RMSE$_{z1}$ (m)[a] | RMSE$_{z2}$ (m)[b] | LoD (m) |
| --- | --- | --- | --- | --- | --- |
| DoD1 | RTK/PPK drone ("Slow") | RTK/PPK drone ("Slow") | 0.08[c] | 0.08[c] | 0.34 |
| DoD2 | LiDAR | RTK/PPK drone ("Slow") | 0.04[d] | 0.08[c] | 0.27 |
| DoD3 | RTK/PPK drone ("Slow") | RTK/PPK drone ("Rapid1") | 0.08[c] | 0.16[e] | 0.54 |
| DoD4 | LiDAR | RTK/PPK drone ("Rapid1") | 0.04[d] | 0.16[e] | 0.49 |
| DoD5 | Non-RTK/PPK drone | Non-RTK/PPK drone | 2.14[f] | 2.14[f] | 9.10 |
| DoD6 | LiDAR | Non-RTK/PPK drone | 0.04[d] | 2.14[f] | 6.43 |

[a]RMSE$_z$ of the pre-disaster DSM.

[b]RMSE$_z$ of the post-disaster DSM.

[c]RMSE$_z$ of "Slow" DSM, as shown in Table 1.

10 [d]RMSE$_z$ of the LiDAR DSM.

[e]RMSE$_z$ of "Rapid1" DSM, as shown in Table 1.

[f]RMSE$_z$ of a DSM generated using a non-RTK/PPK senseFly eBee (no GCPs), from Hugenholtz et al. (2016).

**Figure 1.** Geospatial accuracy results for the "slow" PPK-corrected drone DSM: (a) vertical error histogram with statistics and Shapiro-Wilk (S-W) *p*-value, and (b) DSM of difference, calculated by subtracting $DSM_{LiDAR}$ from $DSM_{drone}$. Blue tints represent elevation overestimations and red tints represent elevation underestimations by $DSM_{drone}$. Buildings with major contiguous DSM differences are boxed in black. The causes of these contiguous DSM differences are due to changes during the 5 years between the LiDAR (2013) and drone (2018) data acquisitions, including new construction (1, 2, 4–10, 12, 14–16), structure removal (3, 5, 11), and parking lot excavation (13).

**Figure 2.** Sample buildings segmented from the dense point cloud (colored by 3D point density), medium-resolution mesh, and high-resolution mesh. Both meshes were generated using identical input imagery and processing settings, except for the mesh resolution setting. Google 3D is shown as a reference for building appearance (Google, 2018).

**Figure 3.** Sample building facades, each represented by a 0.50 m 3D point density raster and a high-resolution mesh segmentation. Red cells within each raster represent data gaps (0 points per cell).

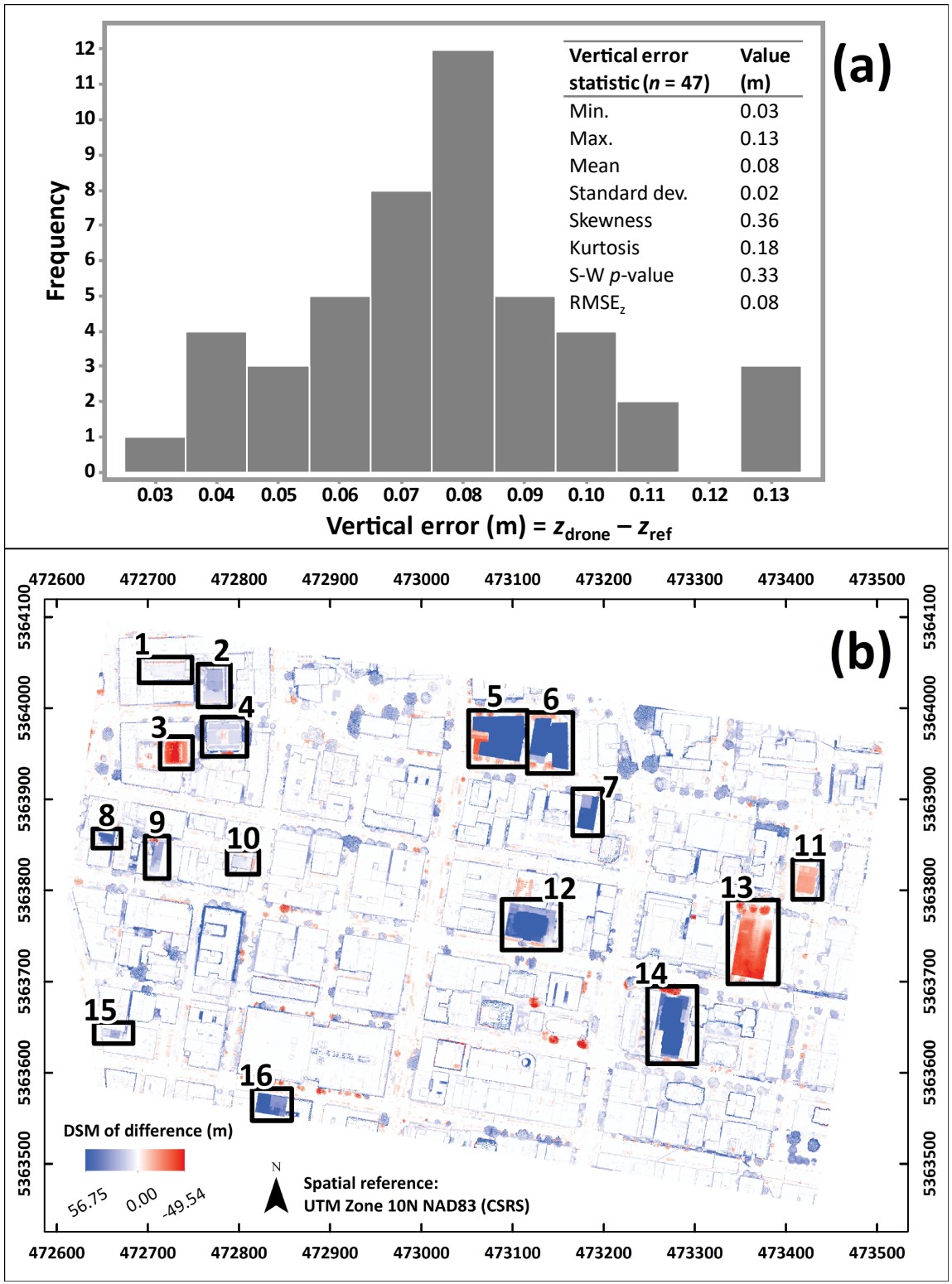

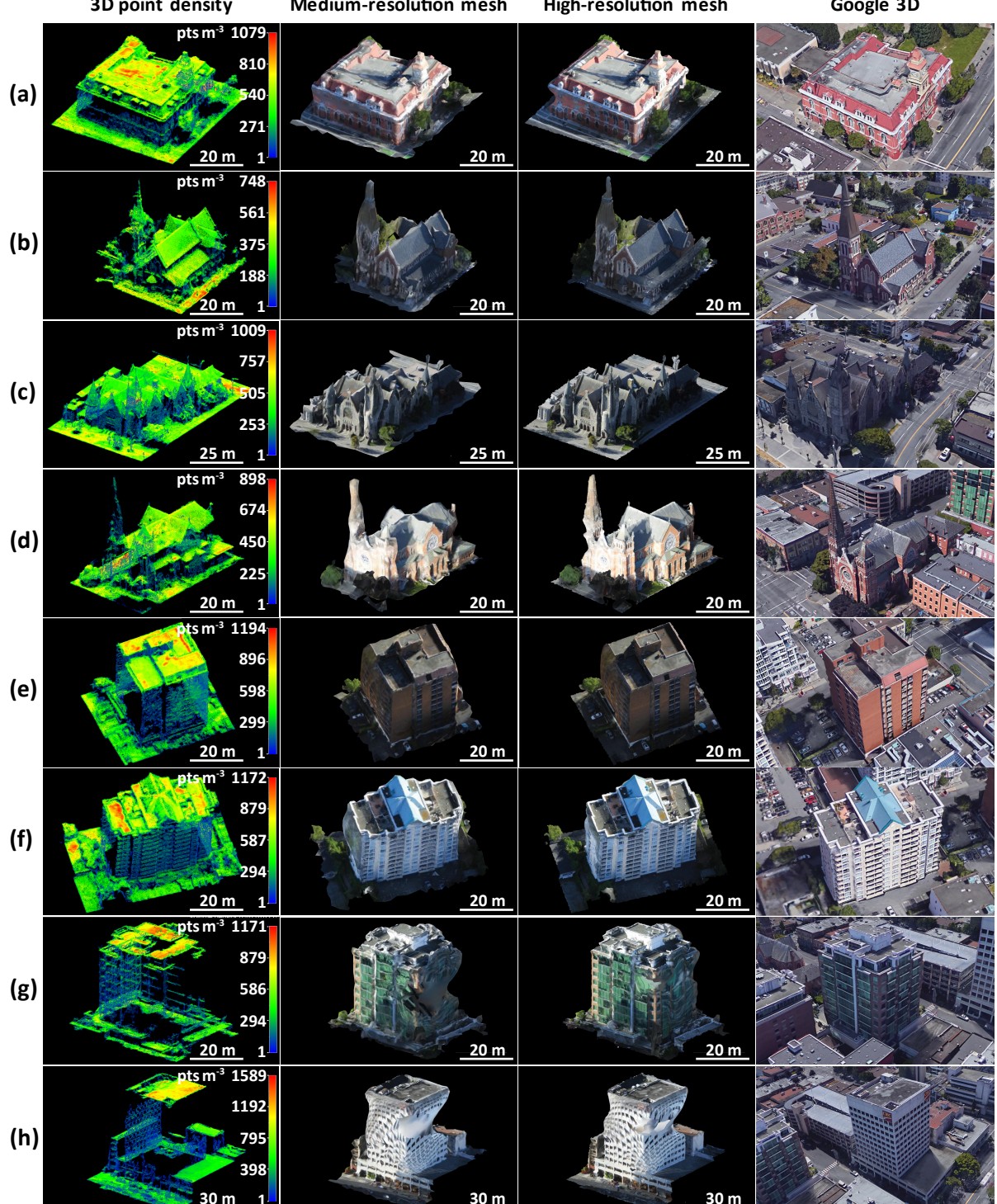

| 3D point density | Medium-resolution mesh | High-resolution mesh | Google 3D |

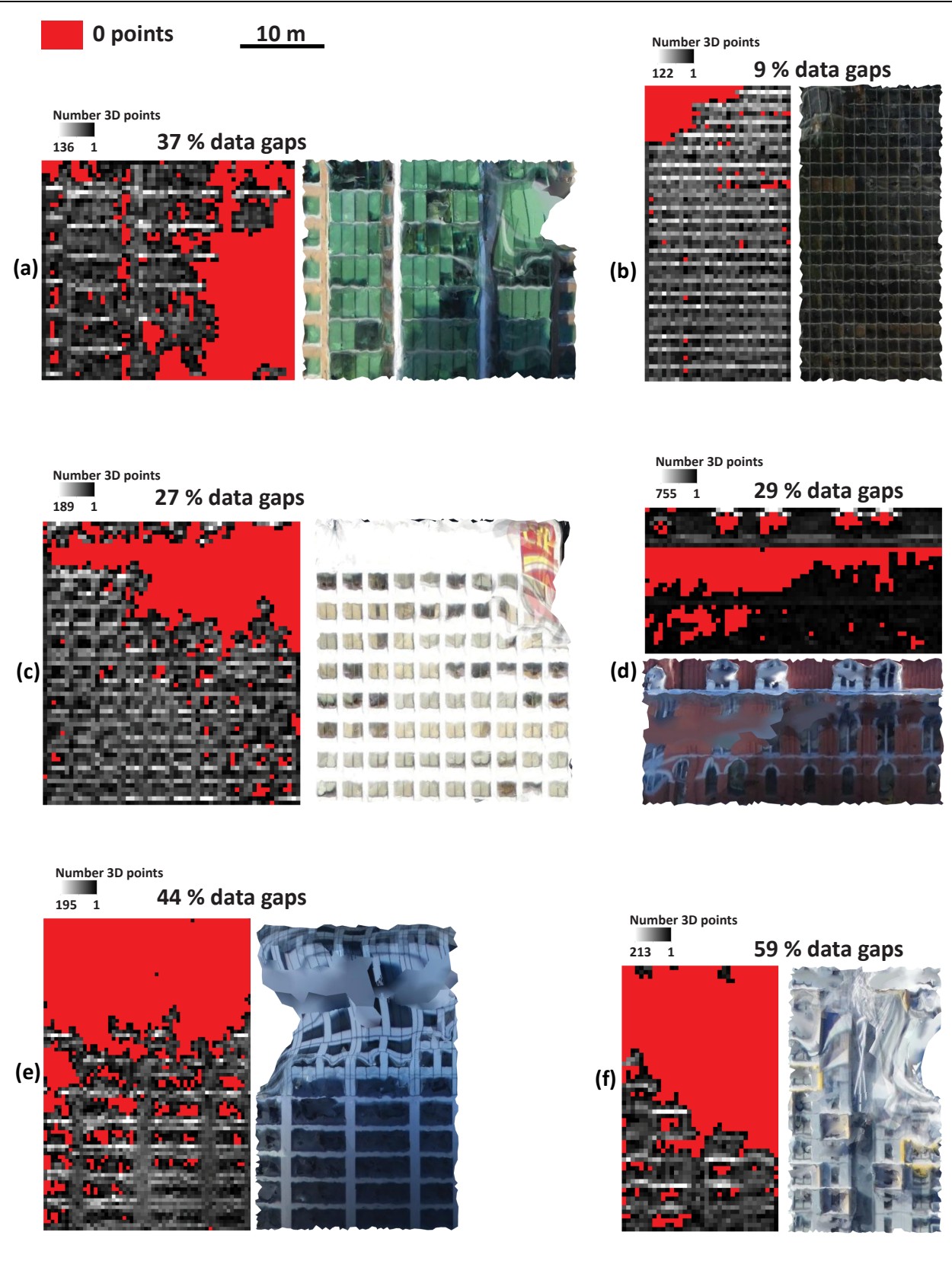

0 points    10 m

**(a)** Number 3D points 136 1 — 37 % data gaps

**(b)** Number 3D points 122 1 — 9 % data gaps

**(c)** Number 3D points 189 1 — 27 % data gaps

**(d)** Number 3D points 755 1 — 29 % data gaps

**(e)** Number 3D points 195 1 — 44 % data gaps

**(f)** Number 3D points 213 1 — 59 % data gaps