# Peer review of "Pre-disaster mapping with drones: an urban case study in Victoria, BC, Canada"

_Natural Hazards and Earth System Sciences, 2018_

## Referee Comment (RC1) · Anonymous Referee #1 · 9 Feb 2019

GENERAL COMMENTS The manuscript presents an interesting study about the potential of terrain elevation data sets and façade images generated from unmanned aerial vehicles (UAVs, also known as drones) to support post-disaster rescue decision making. The study has also a strong practical relevance.

In my opinion, a discussion on the applicability of the proposed data acquisition methods in different conditions from those in Victoria (Canada), e.g. different types of buildings or different city layouts, and also the limitations related to building destruction and weather conditions can impose on the fly-ability of UAVs, should be included in the manuscript.

[Figure]

In general, the manuscript is well written and clear, and the figures and tables are informative and of good quality. Below I suggest a few minor points that the authors may consider to improve the quality of the manuscript

SPECIFIC COMMENTS Page 1, Lines 14-15. This sentence should be rephrased/ improved as it is too general and not completely correct, as it ignores many factors that may minimise the impact of natural hazards in cities (increased quality of construction, alarm systems, proximity to rescue services, . . .).

Page 3, Line 4: according to many guidelines the % symbol should not be preceded by a space. This happens in many other parts of the manuscript. Please consider to revise

Page 3, Line 5: ". . . report. . . conducted. . .". I believe reports do not conduct assessments. Perhaps "present". Please consider to adjust the sentence.

Page 5, Line 2: "GNSS" all acronyms should be defined when they are used for the 1st time in the text to avoid ambiguity. Is GNSS the acronym for "Global Navigation Satellite SystemÂż? Please check other acronyms that are not defined in the manuscript.

Page 6, Line 13: a reference to the software should be added.

Page 6, Line 16: "to" seems to be missing in the sentence

Page 8, Line 1: a "that" seems to be missing in this sentence

Page 8, Line 24: "was assessed going forward"? what do the authors mean with this? Please consider to rephrase the sentence.

Page 9, Line 27: "single story building" instead?

Page 9, Line 32: should read ". . . sub-meter LoDs. . ." instead of sub-decimeter?

Page 10, Line 25: ", . . . but achieve a fraction of time. . .". This part of the sentence is not clear. Please revise.

[Figure]

Figure 2: the font of the 3D point density images legend/scale is very small and difficult to read.

---

## Referee Comment (RC2) · Anonymous Referee #2 · 12 Feb 2019

The use of drones for natural hazards damages evaluation is a well-known topic. It is important to point out that there is a special issue published on NHESS dedicated to UAV and natural hazards, I think that authors can find several interesting suggestions considering the published revision paper or the others. Another revision paper has been published by Gomez, C. and Purdie, H.: UAV- based Photogrammetry and Geo-computing for Hazards and Disaster Risk Monitoring – A Review, Geoenvironmental Disasters, 3, 1–11, 2016. Page 3 line 20: the use of nadiral acquisition (both by drones or planes) can be critical in urbanized areas. In Giordan et al 2018, (see comment on chapter 4.2) the effect of damages caused by a flood was defined using a mixed approach based on drone and terrestrial acquisitions. Page 4 line 15 "The RTK/PPK
image georeferencing capabilities of the drone replaced the need for ground control points (GCPs), which are not practical to distribute and survey in an emergency mapping context." This is not correct. The RTK/PPk correction improves the accuracy of images acquisition points. The number and the needs of GCPs depend on the required accuracy of the SFM results. During the mission planning, it is possible to have an estimation of final accuracy and decide if GCPs are required or not. For fast acquisitions, often GCPs are not required, but for a pre-event acquisition, the required accuracy should be high, and I do not think that it is possible to avoid GCPs. Page 6 line 10: In my experience, this is not the correct way to operate. The first step is the check of the right alignment of surveys. This can be done in particular using large plane areas (like car parking). Then you can compare buildings or other structures. The validation of the right position of DTM is mandatory to be sure that all used DSM are correct form the geodetic point of view. In an exercise like the one presented by authors, they could easily use as a sequence of Ground checkpoints to assure the accuracy of the obtained DSM. These checkpoints can be acquired using natural or artificial elements like (manholes) also after the UAV acquisition. To be rigorous, authors should present a more detailed study of the accuracy of the obtained DSM. This is a crucial point because the accuracy of the DSMs comparison is a function of the accuracy of used DSMs. Chapter 2.6 it is not clear which is the goal of this chapter. Using nadiral images for facades is not correct, and this is not a novelty. Authors should clarify better if the final goal is the identification of damages comparing the geometry of roofs or the study of facades. Authors should present the metadata of 2013 LiDAR before using it as a benchmark like the number of acquired points per meters, the accuracy of the survey, and the density of DSM point cloud. In particular, the DSM density is an important data. If the LiDAR density is not adequate, how authors can be sure that they comparing two points acquired in the same position or they are comparing a surveyed point and an artifact?

Chapter 4.1 the presented "key lesson 1" is quite trivial. Authors presented obvious data for people familiar with LiDAR and drones DSM. Several critical issues are quite

evident in this chapter: the most critical point is the a priori definition of LiDAR resolution and accuracy using García-Quijano et al. (2008). The final resolution of LiDAR surveys is a function of many parameters, like the point density, the flight velocity, the post processing accuracy, and many others. In this paper, authors never mentioned the characteristics of the available LiDAR survey. Another important element is that without the acquisition of checkpoints, authors are not able to define the accuracy of their UAV DSM. I think that this lack of information cannot be accepted in a scientific paper. Page 10, line 5. The presence of differences in the geometry of several houses in the studied area could be useful for better development of the DSM comparison methodology. Using the comparison of DSM an images to check the first results, authors can be able to distinguish damages from building modifications. An improvement of the presented approach and the definition of an effective methodology for the recognition of damages can be an essential add value for this work, and it can also reduce the need of a continuous update of the DSM, which can generate a strong improvement of cost with a limited benefit. The only real result presented in chapter 4.1 is the difference between the results obtained by pix4d using the "rapid" and "full" point cloud. In my opinion, this cannot be considered an adequate result.

Chapter 4.2 the presented "key lesson2" is focused on an interesting point. The nadiral acquisition of an urbanized area is not enough for the correct reconstruction of facades. Giordan et al. (Giordan, D., Notti, D., Villa, A., Zucca, F., Calò, F., Pepe, A., Dutto, F., Pari, P., Baldo, M., and Allasia, P.: Low cost, multiscale and multi-sensor application for flooded areas mapping, Nat. Hazards Earth Syst. Sci., 18, 1493-1516, 2018) published a multi-scale approach aimed to detect and measure damages on facades. The approach is different, but the topic is important for a correct estimation of damages. One of the problems of this article is the organization. If the authors want to analyze facades, they have to introduce this topic in advance and propose a possible methodology. The publication of a sequence of well-known limitations cannot be considered sufficient for an international scientific journal like NHESS.

---

## Author Comment (AC1) · 1 Mar 2019

We sincerely thank Referee #1 for their constructive, helpful, and thorough review that has improved this paper. Below, we address each referee comment. Each referee comment is in **BOLD** and our response directly below. Page/line numbers in referee comments refer to the original submission. Page/line numbers in responses refer to lines in the revised manuscript with Track Changes mode enabled (prior to accepting the changes).

**GENERAL COMMENTS**

1. **The manuscript presents an interesting study about the potential of terrain elevation data sets and façade images generated from unmanned aerial vehicles (UAVs, also known as drones) to support post-disaster rescue decision making. The study has also a strong practical relevance.**

2. **In my opinion, a discussion on the applicability of the proposed data acquisition methods in different conditions from those in Victoria (Canada), e.g. different types of buildings or different city layouts, and also the limitations related to building destruction and weather conditions can impose on the fly-ability of UAVs, should be included in the manuscript.**
   - We thank the referee for this important consideration. We added the following paragraph to the end of Section 4.2: "The data acquisition methods used in this study will need to be adapted to fit the conditions of different urban areas. For example, flight altitude will need to be adjusted to give a safe vertical clearance from the tallest building. If the terrain in the area is sloped, elevation data should be input to the flight planning software to keep the flight altitude constant. A grid of flight lines is recommended, although its orientation and image overlap will vary depending on factors such as building layout and density. In a post-disaster context, a takeoff and landing location may be difficult to locate and access due to widespread destruction. Weather conditions such as high winds and rain following storm events may pose challenges to the flying ability of lightweight drones. Atmospheric conditions such as haze and smoke limit optical sensors in imaging destruction. These factors are examples of considerations that should be made when adapting the data acquisition methodology in this study".

3. **In general, the manuscript is well written and clear, and the figures and tables are informative and of good quality. Below I suggest a few minor points that the authors may consider to improve the quality of the manuscript**

**SPECIFIC COMMENTS**

4. **Page 1, Lines 14-15. This sentence should be rephrased/improved as it is too general and not completely correct, as it ignores many factors that may minimise the impact of natural hazards in cities (increased quality of construction, alarm systems, proximity to rescue services, . . .).**
   - We thank the referee for this important point. On page 1, lines 14-17, we revised the sentence and the proceeding sentence as such: "Increasing global population and urbanization (particularly in vulnerable areas) are factors that can contribute to increased death and destruction by natural hazards like earthquakes and tropical cyclones. In addition to initiatives such as increased quality of construction, alarm systems, and proximity to rescue services, pre-disaster mapping can help increase a city's resilience against disasters (Pu, 2017)".

5. **Page 3, Line 4: according to many guidelines the % symbol should not be preceded by a space. This happens in many other parts of the manuscript. Please consider to revise**

- We followed the NHESS manuscript preparation guidelines (link below) when we decided to include spaces between numbers and units (e.g., %, m, °). The specific guideline is listed under the section "Manuscript composition", subsection "Figure content guidelines", item 4: "Spaces must be included between number and unit (e.g. 1 %, 1 m)". The description for "Figure content guidelines" reads "In order to facilitate consistency with our language and typesetting guidelines applied to the text of the manuscript, please keep the following in mind when producing your figures". Therefore, we interpreted these figure guidelines to be applicable to the text. However, if our interpretation is incorrect, we will remove the spaces between numbers and the % symbol.
- NHESS manuscript preparation guidelines we consulted: https://www.natural-hazards-and-earth-system-sciences.net/for_authors/manuscript_preparation.html

6. **Page 3, Line 5: ". . . report. . . conducted. . .". I believe reports do not conduct assessments. Perhaps "present". Please consider to adjust the sentence.**

- On page 3, line 6, we replaced "conducted" with "presented": "A 2016 report on the seismic vulnerability of Victoria  presented a risk assessment…".

7. **Page 5, Line 2: "GNSS" all acronyms should be defined when they are used for the 1st time in the text to avoid ambiguity. Is GNSS the acronym for "Global Navigation Satellite System"? Please check other acronyms that are not defined in the manuscript.**

- Page 5, lines 10-11: we defined GNSS
- We also defined SODA (page 4, line 17), RGB (page 4, line 17), NRCan (page 4, line 26), and CGG2013 (page 6, line 5).

8. **Page 6, Line 13: a reference to the software should be added.**

- Page 7, line 18: We added an in-text citation to CloudCompare software. We added the citation to the reference list.
- For consistency, we also added in-text citations for senseFly eMotion (page 4, line 28), Pix4D Pix4Dmapper (page 5, line 12), and ESRI ArcMap (page 7, line 9), and added them to the reference list.

9. **Page 6, Line 16: "to" seems to be missing in the sentence**

- Page 6, line 14: We added "to" to the following: "ASPRS (2015) recommend vertical checkpoints to be…".

10. **Page 8, Line 1: a "that" seems to be missing in this sentence**

- Page 8, line 18: We added "that" to the following: "With a 0.31 m average point spacing, it is possible that the LiDAR point cloud…".

11. **Page 8, Line 24: "was assessed going forward"? what do the authors mean with this? Please consider to rephrase the sentence.**

- Page 9, lines 8-9: After considering this comment, we realized this sentence is unnecessary, and have removed it from the text.

**12. Page 9, Line 27: "single story building" instead?**

- We would like to retain "single building story" because the use of DSMs to detect building collapse can include partial collapses such as single-story collapse (Fig. 1) and roof collapse (Fig. 2) within a multi-story building. We believe "single building story" is the more general term, as it includes single-story buildings and partial collapses.

[Figure]

Fig. 1. Single-story collapse. Copied from So (2016).

[Figure]

Fig. 2. Roof collapse. Copied from So (2016).

So, E. (2016). *Estimating Fatality Rates for Earthquake Loss Models.* London: Springer.

**13. Page 9, Line 32: should read ". . . sub-meter LoDs. . ." instead of sub-decimeter?**

- We thank the referee for identifying this error. On page 10, line 19, we changed "sub-decimeter" to "sub-meter".

**14. Page 10, Line 25: ", . . . but achieve a fraction of time. . .". This part of the sentence is not clear. Please revise.**

- Page 11, lines 15-16: We revised the sentence as follows: "Lightweight RTK/PPK-enabled multi-rotors may be more affordable than the senseFly eBee X with SODA 3D camera, but typically have a shorter battery life and subsequently lower areal coverage than fixed wings."

**15. Figure 2: the font of the 3D point density images legend/scale is very small and difficult to read.**

- In Figure 2, we increased the size of the text in the legend and scale.

---

## Author Comment (AC2) · 1 Mar 2019

We sincerely thank Referee #2 for their constructive, helpful, and thorough review that has improved this paper. Below, we address each referee comment. Each referee comment is in **BOLD** and our response directly below. Page/line numbers in referee comments refer to the original submission.

1. **The use of drones for natural hazards damages evaluation is a well-known topic. It is important to point out that there is a special issue published on NHESS dedicated to UAV and natural hazards, I think that authors can find several interesting suggestions considering the published revision paper or the others. Another revision paper has been published by Gomez, C. and Purdie, H.: UAV- based Photogrammetry and Geocomputing for Hazards and Disaster Risk Monitoring – A Review, Geoenvironmental Disasters, 3, 1–11, 2016.**

   - We thank the referee for recommending these publications. We performed an extensive review of the literature and referenced what we believe to be the most relevant and applicable papers to our study.

2. **Page 3 line 20: the use of nadiral acquisition (both by drones or planes) can be critical in urbanized areas. In Giordan et al 2018, (see comment on chapter 4.2) the effect of damages caused by a flood was defined using a mixed approach based on drone and terrestrial acquisitions.**

   - We thank the referee for providing this context. In Section 4.2 of the original submission, we acknowledged that the addition of terrestrial images to drone-based aerial imagery has been shown to improve the 3D textured mesh model in urban study areas, and referred to Wu et al. (2018) as a demonstrative study. The scope of our study was to investigate the sole use of drone-based aerial imagery, and this imagery set was not nadir. As noted in Section 2.3, the 828 images were collected at an average pitch angle of 7 degrees off nadir.
   - To provide more detail about the obliquity of the imagery set, we propose the following modification to the last sentence of Section 2.3: "A total of 828 oblique images were captured. The median image pitch angle was 7.35 ° off nadir (3.55 ° interquartile range), with a minimum and maximum of 1.22 ° and 11.83 °, respectively".

3. **Page 4 line 15 "The RTK/PPK image georeferencing capabilities of the drone replaced the need for ground control points (GCPs), which are not practical to distribute and survey in an emergency mapping context." This is not correct. The RTK/PPk correction improves the accuracy of images acquisition points. The number and the needs of GCPs depend on the required accuracy of the SFM results. During the mission planning, it is possible to have an estimation of final accuracy and decide if GCPs are required or not. For fast acquisitions, often GCPs are not required, but for a pre-event acquisition, the required accuracy should be high, and I do not think that it is possible to avoid GCPs.**

   - We agree with the referee that GCPs can and should be used for pre-disaster mapping to maximize the geospatial accuracy of the data. We retain our position that GCPs are not practical to use for emergency (i.e., post-disaster) mapping. We used PPK corrections instead of GCPs because of the applicability of this georeferencing method to both pre- and post-disaster mapping.

- To communicate the important point the referee has made, we propose the following modification of this paragraph in Section 2.2: "A senseFly eBee Plus drone with real-time kinematic (RTK)/post-processed kinematic (PPK) functionality and senseFly Sensor Optimised for Drone Applications (SODA) red-green-blue (RGB) 20-megapixel camera were used to collect imagery. The RTK/PPK image georeferencing capabilities of the drone replaced the need for ground control points (GCPs), which are not practical to distribute and survey in an emergency (i.e., post-disaster) mapping context. It is important to note that, for pre-disaster mapping, GCPs should be used to maximize geospatial accuracy. Hugenholtz et al. (2016) demonstrate the improvement in DSM vertical accuracy when using a non-RTK senseFly eBee with GCPs compared to an RTK-enabled senseFly eBee without GCPs. For this pre-disaster mapping exercise in downtown Victoria, we chose to use RTK/PPK image georeferencing because this method is also applicable to post-disaster mapping. One of our objectives was to assess the geospatial accuracy of the pre-disaster data, which has implications for the use of RTK/PPK-enabled drones for post-disaster mapping and change detection applications."

4. **Page 6 line 10: In my experience, this is not the correct way to operate. The first step is the check of the right alignment of surveys. This can be done in particular using large plane areas (like car parking). Then you can compare buildings or other structures. The validation of the right position of DTM is mandatory to be sure that all used DSM are correct form the geodetic point of view. In an exercise like the one presented by authors, they could easily use as a sequence of Ground checkpoints to assure the accuracy of the obtained DSM. These checkpoints can be acquired using natural or artificial elements like (manholes) also after the UAV acquisition. To be rigorous, authors should present a more detailed study of the accuracy of the obtained DSM. This is a crucial point because the accuracy of the DSMs comparison is a function of the accuracy of used DSMs.**
   - We thank the referee for making this critical suggestion. We agree with this comment and propose a modification to the vertical accuracy assessment. We propose to replace the LiDAR checkpoints with 47 ground-surveyed (total station) checkpoints located on sewer manhole covers throughout the study area. These ground checkpoints follow the guidelines for vertical checkpoints as outlined in the 2015 ASPRS Positional Accuracy Standards for Digital Geospatial Data (ASPRS, 2015).
   - The proposed modification of the vertical accuracy assessment will be reflected in changes to Section 2.5, Section 3.1, Section 4.1, Figure 1, Table 1, and Table 2.

5. **Chapter 2.6 it is not clear which is the goal of this chapter. Using nadiral images for facades is not correct, and this is not a novelty. Authors should clarify better if the final goal is the identification of damages comparing the geometry of roofs or the study of facades.**
   - To clarify our goal, we added the following text to the beginning of Section 2.6: "In addition to geospatial accuracy, we wanted to assess the quality of building representation in the drone-derived 3D data. This assessment would have implications on the usability of the 3D data for identifying damages to building roofs and facades".
   - As described in our response to Comment #2, the imagery set was not nadir.

6. **Authors should present the metadata of 2013 LiDAR before using it as a benchmark like the number of acquired points per meters, the accuracy of the survey, and the density of DSM point cloud. In particular, the DSM density is an important data. If the LiDAR density is not adequate, how authors can be sure that they comparing two points acquired in the same position or they are comparing a surveyed point and an artifact?**

   - Please refer to our response to Comment #4, where we propose to use 47 ground checkpoints instead of the LiDAR checkpoints.

7. **Chapter 4.1 the presented "key lesson 1" is quite trivial. Authors presented obvious data for people familiar with LiDAR and drones DSM. Several critical issues are quite evident in this chapter: the most critical point is the a priori definition of LiDAR resolution and accuracy using García-Quijano et al. (2008). The final resolution of LiDAR surveys is a function of many parameters, like the point density, the flight velocity, the post processing accuracy, and many others. In this paper, authors never mentioned the characteristics of the available LiDAR survey. Another important element is that without the acquisition of checkpoints, authors are not able to define the accuracy of their UAV DSM. I think that this lack of information cannot be accepted in a scientific paper.**

   - We thank the referee for identifying the improvement that should be made to our reference for piloted LiDAR vertical accuracy. We propose to remove García-Quijano et al. (2008) as the reference in Section 4.1 and Table 2. Instead, we propose to use the 47 ground checkpoints to calculate the $RMSE_z$ of the LiDAR DSM generated using the piloted LiDAR data from our study area. We will use the $RMSE_z$ of the LiDAR DSM to modify Section 4.1 and Table 2. Additionally, we will add the LiDAR metadata to Section 4.1.

8. **Page 10, line 5. The presence of differences in the geometry of several houses in the studied area could be useful for better development of the DSM comparison methodology. Using the comparison of DSM an images to check the first results, authors can be able to distinguish damages from building modifications. An improvement of the presented approach and the definition of an effective methodology for the recognition of damages can be an essential add value for this work, and it can also reduce the need of a continuous update of the DSM, which can generate a strong improvement of cost with a limited benefit. The only real result presented in chapter 4.1 is the difference between the results obtained by pix4d using the "rapid" and "full" point cloud. In my opinion, this cannot be considered an adequate result.**

   - If we are understanding this comment correctly, then the referee is suggesting we develop a methodology to distinguish between damaged buildings and modified buildings in the DoD. This is an excellent suggestion that would indeed contribute to reduced costs and time associated with continuously updating a pre-disaster DSM. If our interpretation of the referee's comment is correct, then we believe we do not have sufficient data for the proposed analysis. As shown in Figure 1b, the changes during the 5 years between the LiDAR (2013) and drone (2018) data acquisitions include new construction, structure removal, and parking lot excavation. The buildings that underwent new construction and structure removal could serve as examples of "modified buildings" in the referee's proposed analysis. However, our data lack

examples of "destroyed buildings". Therefore, we believe we cannot perform what we interpret as the proposed analysis. However, we propose the following addition to the end of Section 4.1: "To reduce costs and time associated with continuously updating a pre-disaster DSM, future research should focus on developing methodologies to distinguish between construction-modified and disaster-damaged buildings in a DoD".

9. **Chapter 4.2 the presented "key lesson2" is focused on an interesting point. The nadiral acquisition of an urbanized area is not enough for the correct reconstruction of facades. Giordan et al. (Giordan, D., Notti, D., Villa, A., Zucca, F., Calò, F., Pepe, A., Dutto, F., Pari, P., Baldo, M., and Allasia, P.: Low cost, multiscale and multi-sensor application for flooded areas mapping, Nat. Hazards Earth Syst. Sci., 18, 1493-1516, 2018) published a multi-scale approach aimed to detect and measure damages on facades. The approach is different, but the topic is important for a correct estimation of damages. One of the problems of this article is the organization. If the authors want to analyze facades, they have to introduce this topic in advance and propose a possible methodology. The publication of a sequence of well-known limitations cannot be considered sufficient for an international scientific journal like NHESS.**

- We strongly disagree with the referee's comment. We believe our research is novel because this is the first government-approved drone mapping mission over a major Canadian city. This was a multi-stakeholder effort that included the municipal emergency management office, federal aviation authority, and air traffic control. In their review paper concerning RPAS for natural hazards monitoring and management, Giordan et al. (2018) recommend that future research should "propose faster and automated approaches. In particular during emergencies, the time required for RPAS data set processing is an important element that should be carefully considered". Giordan et al. (2018) also recommend that, "In the following years, it would be desirable to witness the transfer of best practices in the use of RPASs be then from the research community to government agencies (or private companies) involved in the prevention and reduction of impacts of natural hazards. The scientific community should contribute to the definition of standard methodologies that can be assumed by civil protection agencies for the management of emergencies".

- Consistent with the recommendations of Giordan et al. (2018), we present and evaluate a legal and plausible scenario. This is evidenced by our description of the multi-stakeholder coordination (Section 1.2), our use of the only legally approved drone for urban overflight in Canada to date (Section 1.2), our gridded flight plan for efficiency (as opposed to circular flights around individual buildings) (Section 1, Section 2.2), our use of PPK image georeferencing (as opposed to GCPs) (Section 2.2), and our examination of "rapid" image processing (Section 2.4, Section 4.1). By constraining our study to comply with the legal and logistical practicalities of pre- and post-disaster mapping in a major Canadian city, we believe our results have implications on the usability of the regulatory-approved drone for assisting in rescue and damage assessment activities. Specifically, the results inform the federal aviation authority (Transport Canada) of the limitations of this drone and camera configuration, and we suggest an equally safe alternative for legal approval (Section 4.2, Section 5). We also provide evidence-based lessons/best practices for practitioners such as emergency management offices. These

best practices pertain to drone hardware (e.g., tilting cameras for 3D mapping [Section 4.2] and RTK/PPK georeferencing for change detection applications [Section 4.1]), data collection (e.g., takeoff and landing locations [Section 2.3] and up-to-date DSMs [Section 4.1]), and data processing (e.g., "rapid" processing for sub-meter building collapse detection [Section 4.1]).

- Additionally, by revising the accuracy assessment as recommended by the referee, we believe we provide a more rigorous analysis of the drone and LiDAR DSM accuracies.

**References:**

American Society for Photogrammetry and Remote Sensing (ASPRS): ASPRS Positional Accuracy Standards for Digital Geospatial Data, Photogramm. Eng. Remote Sens., 81(3), 1–26, doi:10.14358/PERS.81.3.A1-A26, 2015.

García-Quijano, M. J., Jensen, J. R., Hodgson, M. E., Hadley, B. C., Gladden, J. B. and Lapine, L. A.: Significance of Altitude and Posting Density on Lidar-derived Elevation Accuracy on Hazardous Waste Sites, Photogramm. Eng. Remote Sens., 74(9), 1137–1146, doi:10.14358/PERS.74.9.1137, 2008.

Giordan, D., Hayakawa, Y. S., Nex, F. and Tarolli, P.: Review article: the use of remotely piloted aircraft systems (RPASs) for natural hazards monitoring and management, Nat. Hazards Earth Syst. Sci., 18(4), 1079–1096, doi:10.5194/nhess-18-1079-2018, 2018.

Wu, B., Xie, L., Hu, H., Zhu, Q. and Yau, E.: Integration of aerial oblique imagery and terrestrial imagery for optimized 3D modeling in urban areas, ISPRS J. Photogramm. Remote Sens., 139, 119–132, doi:10.1016/j.isprsjprs.2018.03.004, 2018.

---

## Author Response (AR2)

Dr. Kai Schröter and two anonymous referees
Natural Hazards and Earth System Sciences

Dear Dr. Schröter and Referees,

Please find enclosed our second resubmission containing suggested revisions from one anonymous referee for our article entitled "Pre-disaster mapping with drones: an urban case study in Victoria, BC, Canada" by Maja Kucharczyk and Chris Hugenholtz.

We would like to thank Anonymous Referees #1 and #3 for their review of the first resubmission, and Anonymous Referee #3 for their constructive suggestions for further improving the paper. The referee comments mainly pertained to the applicability of the methodology to different urban areas, and the organization of the paper. We believe that we have sufficiently addressed the comments.

A record of all changes to the manuscript can be found in the attached marked-up manuscript. Below, we address each referee comment. Each referee comment is in **BOLD** and our response directly below. Page/line numbers in referee comments refer to first resubmission. Page/line numbers in responses refer to lines in the marked-up manuscript showing tracked changes.

Thank you for your continued consideration of our article for publication in *Natural Hazards and Earth System Sciences*.

Sincerely,

Maja Kucharczyk and Chris Hugenholtz

**Comments from Anonymous Referee #3:**

**This paper explores the use of a fixed-wing UAV for cityscape mapping and change detections in building shapes. The manuscript has been once revised according to the previous review. The authors demonstrate the use of eBee UAV for the safe and efficient flight over the urbanized area of Victoria, producing an accurate 3D model of buildings. The reviewer carefully read through the revised manuscript and found that the paper is potentially interesting for the readers of the journal NHESS, but also feels that it needs to be further improved to be a scientific paper published in the journal. The major problems correspond to the issues the previous reviewers pointed out in that 1) the applicability of this methodology needs to be further assessed and 2) the organization of the paper needs to be revised.**

**Applicability**

1. **The authors addressed this issue, which was raised by the previous reviewers, by adding a paragraph in the discussion on the potential applicability of their methodology in other cases with differing conditions. However, to the reviewer's point of view, the added description is a bit insufficient and it needs further reconsideration. Although some of the potential differences in the condition for other areas are presented, no word is provided regarding those in legal and local municipal restrictions. However, this is one of the main issues of this study. If the authors emphasize the novelty of this study in performing the UAV flight in the urbanized area of central Victoria, further discussion on the repeatability or applicability of such legal and practical arrangements would be required. The reviewer is aware of some related descriptions of this issue in the other parts of the manuscript, but they could probably be summarized not in a paragraph but as a separate section. The novelty of this study should not be based on the fact of "the first government-approved drone mapping mission over a major Canadian city" (authors' reply to the reviewer #2 comment) but on the originality of its methodological approach and potential applicabilities.**

   - We agree with the reviewer that the discussion of methodological applicability should be expanded with legal aspects. We created a new discussion section, 4.3 Applicability to other urban areas. In this section, we included the original discussion points pertaining to scene characteristics and weather/atmospheric conditions, and added a paragraph about legal considerations that may impact the transferability of the methodology.

**Organization**

2. **Related to the above-mentioned issue, the organization of this manuscript seems necessary to be further improved. Although this has been pointed out by the previous reviewer, the authors did not agree with reorganizing their manuscript. However, still, the present reviewer found many portions of the manuscript disorganized, which make it hard for the manuscript to be straightforward. For instance, as noted below, many parts of the discussion could be presented in methods or results. Accordingly, the current result section does not sufficiently represent all the results.**

   - We thank the reviewer for providing specific suggestions on how to reorganize the paper. We made the changes, as explained below.

3. **P9 L5-13 These descriptions regarding the LoD should be given in the method and result sections.**

- We moved the DoD and LoD descriptions to the methods section 2.5 Geospatial accuracy assessment and results section 3.1 Geospatial accuracy of drone DSM.

4. **P9 L13-15 These are also methods and results.**
   - We moved the DoD and LoD descriptions to the methods section 2.5 Geospatial accuracy assessment and results section 3.1 Geospatial accuracy of drone DSM.

5. **P9 L15-L18 The LIDAR specification should be presented in methods.**
   - We moved the LiDAR specification to the methods section 2.5 Geospatial accuracy assessment.

6. **P9 L21-31 The arguments regarding the DoD and LoD sounds valid, but these are mostly results.**
   - We moved the DoD/LoD arguments to the results section 3.1 Geospatial accuracy of drone DSM. We also briefly discuss the implications of the DoD/LoD analysis in the discussion section 4.1 Key lessons: drone geospatial accuracy and up-to-date, pre-disaster DSMs.

**Some other comments follow:**

7. **P1 L13 Here could be a subsection such as "1.1. Background", so that the "1.1. Objectives" will be "1.2."**
   - P1 L13: We added the subsection title 1.1 Background and renumbered the subsequent subsections.

8. **P7 L26 The explanation of the acronym DoD was already given.**
   - P8 L23: We removed the redundant explanation for the acronym DoD.

9. **P9 L18-21 Why don't the authors use their own non-corrected data? It must be possible to generate a dataset without RTK or PPK corrections from their eBee images.**
   - We thank the reviewer for this recommendation. We agree that this would be a straightforward assessment that would improve the LoD comparison by providing a non-RTK/PPK drone $RMSE_z$ value using the data collected in this study. Unfortunately, we were unable to obtain the original image geotags from the flight operator because they no longer had these data. Therefore, we retained Hugenholtz et al. (2016) as a reference for non-RTK/PPK drone DSM vertical accuracy. We believe the use of this reference value is reasonable, as Hugenholtz et al. (2016) calculated the $RMSE_z$ of a DSM derived from a non-RTK/PPK senseFly eBee using RTK GNSS checkpoints. The $RMSE_z$ from the reviewer's proposed assessment would likely be similarly high to what was obtained by Hugenholtz et al. (2016), which would result in similarly high LoDs, thus retaining the recommendation for RTK/PPK-enabled drones.

10. **P10 L1-9 It seems OK to insist that the up-to-date datasets are necessary for such a changing urban area. However, why don't the authors discuss the use of airborne LIDAR itself as the continuous monitoring approach, even if it is costly? As such, the advantage of UAV mapping could be discussed further.**
    - P11 L10: We thank the reviewer for recommending this additional point of discussion. Our intention was to recommend a continuous monitoring approach regardless of the surveying technique used. Therefore, we did not recommend a specific technique in this paragraph. To emphasize that a variety of techniques can be used, we added the following sentence: "Up-to-date DSMs can be derived from airborne photogrammetry or laser scanning". We believe we cannot discuss cost effectiveness without first performing a per-area cost analysis, which we believe is out of the scope of this research.

[revised manuscript text omitted]